# Space Omics and Tissue Response in Astronaut Skeletal Muscle after Short and Long Duration Missions

**DOI:** 10.3390/ijms24044095

**Published:** 2023-02-17

**Authors:** Dieter Blottner, Manuela Moriggi, Gabor Trautmann, Maria Hastermann, Daniele Capitanio, Enrica Torretta, Katharina Block, Joern Rittweger, Ulrich Limper, Cecilia Gelfi, Michele Salanova

**Affiliations:** 1Institute of Integrative Neuroanatomy, Charité—Universitätsmedizin Berlin, Corporate Member of Freie Universität Berlin, Humboldt-Universität zu Berlin, and Berlin Institute of Health, 10115 Berlin, Germany; 2NeuroMuscular System & Signaling Group, Center of Space Medicine and Extreme Environments, 10115 Berlin, Germany; 3Department of Biomedical Sciences for Health, University of Milan, 20133 Milan, Italy; 4IRCCS Orthopedic Institute Galeazzi, 20161 Milan, Italy; 5Institute of Aerospace Medicine, German Aerospace Center (DLR), 51147 Cologne, Germany; 6Department of Pediatrics and Adolescence Medicine, University Hospital Cologne, 50937 Cologne, Germany; 7Department of Anaesthesiology and Intensive Care Medicine, Merheim Medical Center, Witten/Herdecke University, 51109 Cologne, Germany

**Keywords:** skeletal muscle, proteomics, spaceflight, structural proteins, protein signaling pathways

## Abstract

The molecular mechanisms of skeletal muscle adaptation to spaceflight are as yet not fully investigated and well understood. The MUSCLE BIOPSY study analyzed pre and postflight deep calf muscle biopsies (m. soleus) obtained from five male International Space Station (ISS) astronauts. Moderate rates of myofiber atrophy were found in long-duration mission (LDM) astronauts (~180 days in space) performing routine inflight exercise as countermeasure (CM) compared to a short-duration mission (SDM) astronaut (11 days in space, little or no inflight CM) for reference control. Conventional H&E scout histology showed enlarged intramuscular connective tissue gaps between myofiber groups in LDM post vs. preflight. Immunoexpression signals of extracellular matrix (ECM) molecules, collagen 4 and 6, COL4 and 6, and perlecan were reduced while matrix-metalloproteinase, MMP2, biomarker remained unchanged in LDM post vs. preflight suggesting connective tissue remodeling. Large scale proteomics (space omics) identified two canonical protein pathways associated to muscle weakness (necroptosis, GP6 signaling/COL6) in SDM and four key pathways (Fatty acid β-oxidation, integrin-linked kinase ILK, Rho A GTPase RHO, dilated cardiomyopathy signaling) explicitly in LDM. The levels of structural ECM organization proteins COL6A1/A3, fibrillin 1, FBN1, and lumican, LUM, increased in postflight SDM vs. LDM. Proteins from tricarboxylic acid, TCA cycle, mitochondrial respiratory chain, and lipid metabolism mostly recovered in LDM vs. SDM. High levels of calcium signaling proteins, ryanodine receptor 1, RyR1, calsequestrin 1/2, CASQ1/2, annexin A2, ANXA2, and sarco(endo)plasmic reticulum Ca(2+)-ATPase (SERCA1) pump, ATP2A, were signatures of SDM, and decreased levels of oxidative stress peroxiredoxin 1, PRDX1, thioredoxin-dependent peroxide reductase, PRDX3, or superoxide dismutase [Mn] 2, SOD2, signatures of LDM postflight. Results help to better understand the spatiotemporal molecular adaptation of skeletal muscle and provide a large scale database of skeletal muscle from human spaceflight for the better design of effective CM protocols in future human deep space exploration.

## 1. Introduction

Human spaceflight so far has emphasized space-related health challenges, such as microgravity-induced impaired body functions (microgravity (µG) very small fractional and/or zero net gravitational forces (0G) pull on the unloaded whole body in relation to normal Earth gravity (1G)) by pathophysiological adaptations [1] affecting astronauts’ health management, e.g., with risks of injuries to the musculoskeletal system [2], composed of skeletal muscle and bone as interconnected physiological “executers” to human motions on Earth [3,4,5].

Earlier work focused on fundamental aspects of human skeletal muscle adaptation and maladaptation in spaceflight, providing our current knowledge on microgravity-induced negative effects affecting muscle and bone structure and function, impaired human performance, health risks, and potential exercise benefits of astronauts during spaceflight [6,7,8,9,10]. Novel research priorities on muscle and bone, however, have identified, for the cellular tissue to organ systems level, deeper insights concerning key molecular mechanisms critical to maintain global muscle mass, muscle fiber phenotype, and bone cell and tissue architecture [7,11], signal transmission [12], or mechanostimulation [13] following body unloading conditions in human space life science [14]. 

In human muscle research, the use of small tissue biopsies from a local muscle of interest is an expedient way to study adaptation and maladaptation and their underlying molecular mechanisms straight in a given skeletal muscle under experimental study conditions [15,16,17] but also in astronauts before and after spaceflight [18,19]. This well-established approach enables further elucidation of the still not fully understood disuse-induced molecular adaptation in skeletal muscle and its consequences in terms of modality and design optimization of future exercise protocols as countermeasure to musculoskeletal atrophy to combat impaired performance control in spaceflight.

A recent pilot study (op. nom. SARCOLAB) on astronauts of the International Space Station (ISS) reported functional calf muscle decrements with variable although mostly negative outcomes in up or down leveling of muscle-specific parameters, but also in single fiber mechanics, costameric proteins, and proteome changes (contractile vs. anerobic/aerobic, energy transduction metabolism), suggesting variable inflight exercise benefit outcomes reported by two crewmembers [20]. 

The present work is part of the MUSCLE BIOPSY (op. nom.) experiment on several more ISS astronauts as study participants. We report (i) on structural tissue phenotype adaptation (myofiber size and type), and (ii) on the muscle proteome (space omics) responses found in pre vs. postflight biopsy samples obtained from the deep calf soleus (SOL) muscle, one typical postural deep calf leg reference muscle among other antigravity muscles of the human body (e.g., quadriceps, muscles of the back) considered highly responsive to unloading conditions on Earth and in Space [21]. A major study aim was to (i) document molecular muscle adaptation in long-duration mission (LDM) astronauts, and (ii) to provide a more detailed analyses of changes in muscle tissue, muscle contraction proteins, and changes in several canonical and muscle-specific key metabolic pathways (i.e., calcium signaling, lipid metabolism, transport and high energy, stress and other proteins). The SOL muscle datasets were further analyzed by functional category of proteins using ingenuity pathway analysis (IPA) to highlight acute space omics responses in a short-duration mission (SDM) astronaut (11 days spaceflight) without routine exercise sessions vs. chronic space omics responses in several other LDM astronauts (six months/180 days on ISS) performing routine inflight exercise countermeasure sessions (up to 2.5 h/d) throughout their extended mission stay on ISS [22].

## 2. Results

### 2.1. Anthropometric Data

Both short-duration mission (SDM) and long-duration mission (LDM) astronauts A–D included in this study showed comparable body mass index (BMI) from 23.5 to 25.8 before launch to Space (Table 1). 

### 2.2. Training Protocol

Unlike for SDM, the LDM astronauts A–D participated in a routine onboard countermeasure (CM) exercise regime every week during their stay on the ISS (Sundays off) using either of the CM exercise devices at their own discretion (CEVIS/T2/ARED) per each flight day (FD) session as suggested by the Medical Operations team (MedOPs) of the astronauts’ affiliated Space Agency. The total FD number of onboard exercise for each of the CM devices are listed to show compliance to inflight exercise protocols (Table 2). In particular, ARED CM session days (device usage during session days) were variable between astronauts, e.g., more days for astronaut A (107 days ISS), or fewer days with astronaut C (29 days ISS), compared to nearly similar numbers of CEVIS/T2 CM exercise session days for LDM astronauts A–D.

### 2.3. Myofiber Size (Cross-Sectional Area, CSA) Analysis

The pre/postflight SOL myofiber size (CSA) changes were analyzed by histomorphometry and immunohistochemistry for SDM and LDM astronaut muscle biopsies (Figure 1). 

The LDM astronauts revealed moderate rates of myofiber CSA reduction (approximately 25% off post vs. pre), suggesting moderate SOL muscle atrophy rates vs. baseline likely due to their numerous daily inflight CM exercise sessions onboard the ISS (Figure 1A,B). However, the SDM astronaut showed a more severe change of SOL myofiber CSA rate (>50% off post vs. pre) as a clear sign of extensive and rapid myofiber atrophy seen just after a relatively short exposure to microgravity (11 days) in the absence of inflight CM exercise onboard ISS (Figure 1C). 

### 2.4. Histology and Extracellular Matrix Immunomarkers

Hematoxilin-Eosin (H.E) stained scout cryosections from LDM revealed changes in normal muscle tissue histology post vs. preflight (Figure 2). The densely packed cross-sectioned SOL myofibers seen under normal baseline condition (LDM preflight) were found as more loosely packed smaller or larger myofiber groups separated by enlarged and visible tissue spaces resembling intramuscular connective tissue property changes (LDM postflight). The pre/postflight expression of collagen types 4 and 6 were semiquantified by using laser confocal microscopy and immunohistochemistry (Figure 2). Immunomarkers Col4 and Col6 clearly showed reduced immunosignals preflight/(upper panel) vs. postflight (lower panel). Two other ECM molecules associated to regular endomysial connective tissue and the myofiber basal lamina, Perlecan and MMP2, also showed reduced immunosignals in the postflight SOL muscle (Figure 2A). Statistic quantification analysis of immunosignals confirmed reduced or even unchanged mean signal intensities in crewmembers. However, increased signal intensity of Perlecan was detected in one LDM crewmember (Figure 2B).

### 2.5. Proteomic Profile in Acute SDM (9/11 Days ISS Mission) and Chronic LDM Exposure (>6 Months ISS Mission) to Microgravity

Pre/postflight protein levels from *SOL* muscle extracts were analyzed in a total of 5 astronauts, one SDM (acute exposure) and four LDM astronauts (chronic exposure) to microgravity in spaceflight. The entire list of proteins differentially expressed in all comparisons, together with statistical analyses, protein accession number, gene name, and label free data, is available in Appendix A.

Changes in protein abundance were assessed by liquid chromatography (LC) coupled to electrospray tandem mass spectrometry (LC–ESI–MS/MS) and label-free quantification. Out of 1822 identified proteins, the paired Student’s *t*-test (n = 1, SDM; n = 4, LDM; *p*-value < 0.05) revealed 83 proteins changed in LDM astronauts and 123 proteins changed in the SDM astronaut. Identification data from LC–ESI–MS/MS of changed proteins/proteoforms are shown in Appendix A. In Figure 3 the Venn diagram indicates 41 common proteins changed in both LDM and SDM (29 with the same trend and 12 with opposite trend). Moreover, 42 proteins were changed in LDM only, while 82 proteins were changed in SDM. 

By processing data sets using the ingenuity pathway analysis (IPA) software, the identification of canonical pathways of SOL in SDM vs. LDM was achieved. The canonical pathway analysis enables recognition of key signaling associated with differentially expressed proteins. A total of eight pathways were changed significantly in LDM vs. SDM exposure (Fisher’s right-tailed exact test *p*-value < 0.05 and z-score ≥ +2 or ≤−2). Negative and positive values indicate inhibition and activation respectively. As reported in Table 3, two identified pathways (oxidative phosphorylation and TCA Cycle) were inhibited in both conditions even though the oxidative phosphorylation appears nearly recovered upon LDM exposure. Fatty acid β-oxidation, ILK, RHOA, and dilated cardiomyopathy signaling characterize LDM exposure whereas two other pathways (necroptosis and GP6 signaling) characterize SDM exposure.

#### 2.5.1. Structural/Contractile Proteins and Proteins Involved in Calcium Signaling 

Figure 4A shows in detail protein changes according to extracellular matrix (ECM), cytoskeletal, and microtubule organization. Results indicated no changes or few changes in LDM protein levels except a slight down-regulation for plectin (PLEC) and dihydropyrimidinase-related protein 3 (DPYSL3) both involved in cytoskeleton organization; while tubulin beta-4B chain (TUBB4B), involved in microtubule organization, is the only increased protein after LDM exposure to microgravity.

Conversely, the SDM astronaut showed a dysregulation of ECM and of cytoskeletal and microtubule organization with increased levels of collagen alpha-1(VI) chain (COL6A1), collagen alpha-3(VI) chain (COL6A3), lumican (LUM), and fibrillin-1 (FBN1), and decreased levels of microtubule-associated protein (MAP4) and tubulin beta chain (TUBB) involved in the microtubule organization. Concerning cytoskeletal organization: PLEC, alpha actin (ACTA1), Filamin-A (FLNA), moesin (MSN), PDZ LIM domain protein 3 (PDLIM3), synemin (SYNM), vinculin (VCL), and vimentin (VIM) increased. Regarding the inner mitochondrial membrane organization and endoplasmic reticulum (ER): MICOS complex subunit MIC60 (IMMT) and reticulon-4 (RTN4) were slightly decreased under SDM condition in spaceflight. 

Figure 4B shows protein changes related to muscle development, contractile, sarcomeric and tubule organization. A common trend was observed for proteins involved in muscle development such as four and a half LIM domains protein 1 (FLH1) and tripartite motif-containing protein 72 (TRIM72), and of slow type fibers like myosin regulatory light chain 2 (MYL2), myosin light chain 3 (MYL3) and myosin light chain 6B (MYL6B) which decreased both in LDM and SDM muscle biopsies. Myosin-7 (MYH7) and actinin-2 (ACTN2) behaved at variance, with a slight increment in SDM and decrease in LDM exposure. In LDM, myosin-binding protein C, slow-type (MYBPC1) and cofilin-2 (CFL2) decreased while nebulin (NEB), was slightly increased. 

SDM was also characterized by increase of EH domain-containing protein 2 (EHD2), while ankyrin repeat domain-containing protein 2 (ANKRD2), FLH1 and of cysteine-rich domains protein 1 (LCMD1) decreased as troponin I (TNNI1). Proteins related to fast type fibers: myosin light chain 1/3 (MYL1), myosin regulatory light chain 2 (MYLPF), and myosin-2 (MYH2), were significantly increased in SDM, as were proteins involved in the sarcomere organization, e.g., myomesin-2 (MYOM2), tropomyosin beta chain (TPM2), and titin (TTN). 

Figure 5 shows proteins involved in calcium signaling. Protein levels were influenced both in SDM or LDM. Calsequestrin-1 (CASQ1), ryanodine receptor 1 (RYR1), calcium-binding mitochondrial carrier protein Aralar1 (SLC25A12), and cAMP-dependent protein kinase catalytic subunit alpha (PRKACA) increased in LDM whereas protein S100-A1 (S100A1) and sarcalumenin (SRL) levels decreased. Synaptophysin-like protein 2 (SYPL2), from the tubule organization, was at variance, decreased in SDM and increased in LDM muscle biopsies.

In SDM, CASQ1, calsequestrin-2 (CASQ2), RYR1, sarcoplasmic/endoplasmic reticulum calcium ATPase 1 (SERCA) (ATP2A2), and annexin A2 (ANXA2) increased. These results suggest that dysregulation of calcium handling in muscle is a likely signature of microgravity exposure.

#### 2.5.2. Change in Metabolic Protein Abundance According to Glycolysis, TCA Cycle and Mitochondrial Respiratory Chain 

In LDM muscle biopsies, L-lactate dehydrogenase B chain (LDHB) and alpha-enolase (ENO1) decreased (Figure 6A).The SDM muscle biopsy was, however, characterized by a decrement of UTP-glucose-1-phosphate uridylyltransferase (UPG2), glycogen (starch) synthase (GYS1), glycogen debranching enzyme (AGL), fructose-bisphosphate aldolase A (ALDOA), triosephosphate isomerase (TPI1), LDHB, and GPDL1; while ATP-dependent 6-phosphofructokinase (PFKM), pyruvate kinase (PKM), L-lactate dehydrogenase A chain (LDHA), and GPD1 increased. 

Figure 6B shows results from proteins involved in TCA cycle and 2-oxoglutarate pathway. A slight decrement was observed in LDM for PDHA1, IDH2, OGDH, DLST, DLD, FH and GOT2, whereas SLC25A11 was slightly increased. The SDM condition is characterized by a decrease of the TCA cycle enzymes (PDHA1, CS, ACO2, IDH2, OGDH, SUCL2, SDHA and FH) and of enzymes from the oxoglutarate pathway (MDH1, GOT1 and SLC25A11). 

Figure 6C shows changes in the mitochondrial respiratory chain in LDM condition. UQCRC1, UQCRC2, UQCRQ, and MT-CO2 were slightly increased. Concerning complex V, a slight decrement was observed for ATP5A1, ATP5B, ATP5D, ATP5H, ATP5J, and electron transfer flavoprotein subunit (ETFA, ETFB, ETFDH). 

In SDM, seven subunits of complex I, NADH ubiquinone oxidoreductase (NDUFA5, NDUFB8, NDUFB10, NDUFS1, NDUFS2, NDUFS5, NDUFS8) were decreased as cytochrome bc1 complex, UQCRC1 and UQCRC2 of complex III and cytochrome c oxidase, COX4l1, COX5B, and COX6B1 of complex IV, ATP synthase complex, ATP5A1, ATP5B, ATP5C1, ATP5D, ATP5H, ATP5J, and ATP5J2 of complex V decreased. Cytochrome C (CYCS) and CDGSH iron-sulfur domain-containing protein 1 (CISD1) were also decreased.

#### 2.5.3. Lipid Metabolism

Proteins involved in fatty acid β-oxidation are shown in Figure 7. Under LDM conditions, Trifunctional enzyme subunit alpha (HADHA), medium-chain specific acyl-CoA dehydrogenase (ACADM), hydroxyacyl-coenzyme A dehydrogenase (HADH), and 3-ketoacyl-CoA thiolase (ACAA2) decreased. In both SDM and LDMl, Long-chain-fatty-acid—CoA ligase 1 (ACSL1), Very long-chain specific acyl-CoA dehydrogenase (ACADVL), 2,4-dienoyl-CoA reductase (DECR1), delta(3,5)-Delta(2,4)-dienoyl-CoA isomerase (ECH1), enoyl-CoA delta isomerase 2 (ECI2), and enoyl-CoA hydratase (ECHS1) decreased. In SDM only, acetyl-CoA acetyltransferase (ACAT1) and acyl-coenzyme A thioesterase 1 (ACOT1) decreased. 

In LDM, fatty acid α-oxidation: retinal dehydrogenase 1 (ALDH1A1) and delta-1-pyrroline-5-carboxylate dehydrogenase (ALDH4A1) were decreased.

In addition, there are several proteins involved in other lipids, secondary metabolites, amino-acid degradation, and cofactor of biosynthetic pathways, e.g., NADH-cytochrome b5 reductase 3 (CYB5R3), prostaglandin E synthase 2 (PTGES2), alanine aminotransferase 1 (GPT), and ubiquinone biosynthesis protein (COQ9), which were decreased after LDM spaceflight. In acute exposure, NADH-cytochrome b5 reductase 1 (CYB5R1) and lactoylglutathione lyase (GLO1) decreased while epoxide hydrolase 1 (EPHX1) increased.

#### 2.5.4. Behavior of Transport, High-Energy Phosphate Interconversion, Stress and Others Proteins

Figure 8A indicates a change in transport proteins. In LDM, fatty acid-binding protein (FABP3) decreased, whereas voltage-dependent anion-selective channel (VDAC1 and VDAC3) increased.

In SDM, apolipoprotein A-I and A-II (APOA1, APOA2), FABP3, vitamin D-binding protein (GC), myoglobin (MB), serotransferrin (TF), VDAC1, VDAC2, and VDAC3 decreased; while band 3 anion transport protein (SLC4A1) increased.

Figure 8B shows results from proteins involved in high-energy phosphate interconversion. Creatine kinase B-type (CKB) and ADP/ATP translocase 1 (SLC25A4) decreased in both conditions, while adenylosuccinate synthetase isozyme 1 (ADSSL1) was at variance, with a slight increase in LDM and decrement in SDM exposure. LDM was characterized also by decreased levels of creatine kinase S-type (CKMT2), Rab GDP dissociation inhibitor beta (GDI2) and nucleoside diphosphate kinase B (NME2), while increased levels were observed for NAD(P) transhydrogenase (NNT). In SDM, adenylate kinase isoenzyme 1 (AK1) increased, while creatine kinase M-type (CKM) decreased. 

Figure 8C shows protein changes in response to oxidative stress, blood coagulation and acute phase response. Glutathione S-transferase omega-1 (GSTO1) and 60 kDa heat shock protein (HSPD1) decreased in both conditions. Carbonic anhydrase 3 (CA3), peroxiredoxin-1 (PRDX1), thioredoxin-dependent peroxide reductase (PRDX3) and superoxide dismutase (Mn)(SOD2) decreased in LDM exposure only. In SDM exposure, heat shock protein beta-6 (HSPB6), haptoglobin (HP) and antithrombin-III (SERPINC1) were decreased; while peptidyl-prolyl cis-trans isomerase A (PPIA), fibrinogen alpha chain (FGA), and fibrinogen beta chain (FGB) increased. 

Figure 8D shows results from transcription regulation, protein biosynthesis, regulation of apoptotic process and proteasome complex. In LDM, eukaryotic translation initiation factor 5A-1 (EIF5A), Bcl-2-like protein 13 (BCL2L13), proteasome subunit alpha type-7 (PSMA7), glutamine amidotransferase-like class 1 domain-containing protein 3 (GATD3) and SH3 domain-binding glutamic acid-rich protein (SH3BGR) decreased while protein-cysteine N-palmitoyltransferase HHAT-like protein (HHATL) and prohibitin (PHB) increased. In SDM, elongation factor Tu (TUFM), Y-box-binding protein 3 (YBX3), and hydroxysteroid dehydrogenase-like protein 2 (HSDL2) decreased. Conversely, carboxymethylenebutenolidase homolog (CMBL) increased. Other proteins include: ferritin heavy chain (FTH1) increased in both conditions, whereas 14-3-3 protein zeta/delta (YWHAZ) was at variance, with a slight decrement in LDM and an increment in SDM exposure to spaceflight. 

## 3. Discussion

Here, we report on the structural tissue adaptation and on the space omics responses investigated in tissue biopsies from the deep calf soleus muscle (SOL) from International Space Station (ISS) astronauts harvested before (BDC-60 days to launch) and after (R+0/1 day of return) short- (SDM, 11 days), and long-duration missions (LDM, >6 months) from at least five male ISS crewmembers who participated to the MUSCLE BIOPSY experiment. 

In general, considerable differences were observable at the proteomic level between SDM and LDM astronauts especially regarding expression indices of many muscle specific physiological processes, subcellular compartments, and signal transduction pathways. 

While acute microgravity exposure in the SDM group stress the upregulation of two well-known muscle weakness-associated pathways, necroptosis and GP6 signaling, the same effects were less intense or even at variance in the LDM group following chronic exposure to microgravity. These changes could very well be the result of skeletal muscle adaptation to chronic microgravity exposure (approximately six months in space), or alternatively, to a change of course or trend related to the deconditioning and accumulation of various other processes incompletely understood or still not yet explored. Surprisingly, we identified pathways such as the TCA cycle consistently showing the same trends, although less intense, between acute (SDM) and chronic (LDM) microgravity exposure. By contrast, other pathways clearly showed opposite trends, such as oxidative phosphorylation, necroptosis signaling, GP6 signaling in SDM, or fatty acid oxidation, ILK and RHOA signaling, and cardiomyopathy signaling pathway in LDM.

In particular, we found a change in at least eight canonical pathways associated to differentially expressed proteins in all astronaut muscle samples. These were the two SDM-associated pathways, necroptosis, GP6 signaling, and the four LDM-associated pathways, fatty acid β-oxidation, ILK, RHO, and dilated cardiomyopathy signaling that characterize metabolic SOL muscle changes in short vs. long duration mission conditions in spaceflight. Apart from TCA cycle inhibition found under both SDM or LDM conditions, oxidative phosphorylation pathway proteins, most of them known as molecular markers of mitochondrial response in muscle wasting and other disuse scenarios [23], as well as in physical exercise [24], almost recovered following LDM vs. SDM as most likely a signature of inflight CM exercise benefit for the slow-type myofiber-containing SOL during LDM spaceflight. By contrast, the SDM-associated change in necroptosis signaling pathway-associated proteins are known signatures associated to muscle weakness and wasting [25], autophagy/mitophagy [25,26,27], or even aging [28] in disuse conditions or muscle disease. Increased GP6 signaling pathway protein levels similar to those found in SDM were also reported for re-innervation mechanisms of the mammalian (rat) neuromuscular system [29], suggesting possible microgravity-induced acute neuromuscular changes in SDM, however not seen in LDM, that require further investigation.

Impaired intracellular calcium release via the modulation of intracellular calcium release channels, such as RyRs 1 and 3 and a set of related calcium (Ca^2+^)-binding proteins from the specialized intracellular calcium, store longitudinal sarco(endo)plasmic reticulum cisternae required for excitation-contraction coupling [30] in muscle by calsequestrins (CASQ) and other calcium binding proteins is a contributing factor in striated muscle health and disease [31], including clinical muscle fatigue [32]. Altered calcium signaling key protein levels following SDM spaceflight found in this work thus highlights maladaptation of the mostly slow-twitched muscle fiber (type I) containing SOL (expressing both CASQ1 and 2) where an increase in CASQ1 and CASQ2 and two other related proteins (ANXA2, ATP2A1) was detected at variance in both SDM and LDM samples, suggesting (Ca^2+^)-binding proteins as a protein hub to acute microgravity exposition of human muscle (SOL) in spaceflight. The reduction in muscular synaptophysin-like protein 2 (SYPL2), a presynaptic biomarker, recently shown to be important for cachectic mice [33] but also found in an SDM muscle sample compared to LDM, suggests an altered transverse (t)-tubules/sarcoplasmic membrane terminal cisternae communication as a likely early event possibly affected during exposure to µG. Further investigations, however, are required to further support such hypothesis.

As expected, the metabolic protein abundance associated to glycolysis, TCA cycle, mitochondrial respiratory chain, but also to complex I-V protein subunits by functional category analysis was variably changed in LDM, however consistently found as different patterns from the same SDM muscle proteins. At this point, we propose exercise-driven mechanisms (via inflight CM) to be partly responsible for metabolic protein expression triggered by LDM conditions. Similar findings, e.g., from other exercise-driven aerobic, anaerobic, or energy transduction muscle protein changes were found in a previous long-term bedrest study [34] and in the two LDM astronauts of SARCOLAB [20] which are consistent with our present results.

Impaired lipid oxidation induced ectopic fat deposition and lipotoxicity in human skeletal muscle [35] are both hallmarks in sarcopenia [36], whereas exercise promotes a change in anabolic pathways, such as muscular protein synthesis and lipid metabolism, also known as metabolic fueling (via lipid droplets), during physical exercise [37]. In long-term bedrest, as an analog to spaceflight, lipid metabolism functional gene clusters (transcripts) are highly triggered through resistive (RE) and resistive plus vibration exercise (RVE) compared to bed rest-only control, suggesting that exercise-driven support of the two physical exercise modalities potentially affects anabolic metabolism as a benefit outcome following prolonged disuse conditions, such as bed rest [38] or spaceflight. The murine SOL (normal mice) shows greater intramyocellular lipid droplet accumulations compared to their mostly fast-type EDL muscle antagonist [39]. Though we did not study myocellular lipid droplet accumulation [40] in our astronaut SOL biopsies (tissue constraints, small sample size), we conclude that different lipid metabolism patterns detectable in LDM (exercise) samples vs. SDM (no exercise) via heatmap fatty acid alpha/β-oxidation protein expression are likely associated to inflight CM exercise compared to SDM with probably less muscle energy expenditure in spaceflight.

Finally, transport proteins as well as high energy phosphate interconversion, stress, and other proteins showed variable expression patterns under LDM vs. SDM. As an example, oxidative stress proteins levels, e.g., PRDX1, PRDX3, or SOD 2, were decreased in LDM vs. SDM. Oxidative stress induced muscle protein changes were also reported from clinical critical illness [41] and apparently healthy participants in bed rest [42,43]. However, to better understand changes in muscle proteins potentially affected by disuse-induced oxidative stress and its consequences for human spaceflight more targeted omics approaches are required in future work including for example the impact of vitamins and other nutritional interventions related to spaceflight [44,45,46,47]. Thus, the present space omics database obtained from astronaut SOL (pre/postflight) show a high variety in protein expression patterns very different from SDM vs. LDM condition suggesting a strong impact of acute vs. chronic microgravity exposition in spaceflight on skeletal muscle structure and function. The general and more specific proteome study outcome provides yet more evidence of a robust benefit in terms of molecular pathway adaptation in human skeletal muscle plasticity mechanisms observed in LDM astronauts performing routine inflight exercise as countermeasure during their extended stay on the ISS. 

LDM astronauts aboard the ISS usually perform routine inflight CM exercises (up to 2.5 h daily) [22] using three currently available inflight CM devices (CEVIS, T2, ARED) as part of personalized session protocols (at own discretion). Thus pre/postflight SOL myofiber CSA values in LDM astronauts from MUSCLE BIOPSY experiment were different among subjects at least shown after return to Earth. From LDM crewmembers A–D of the MUSCLE BIOPSY study, we found highest pre vs. postflight myofiber CSA change in B, medium change in C and D, and the lowest change in A, thus confirming larger intersubject variability in inflight exercise outcome related to SOL muscle size phenotype adaptation. Similar discrepancies were reported also for the whole calf triceps muscle (gastrocnemius, soleus) structural parameters in the two astronauts of SARCOLAB [20]. We postulate that the magnitude of myofiber CSA loss in an astronaut during or after a spaceflight mission is likely dependent at least in part also on the actual myofiber size acquired and built-up during individual preflight body preconditioning (preflight baseline), and that inflight CM exercise is able at least to support SOL myofibers against major structural maladaptation in spaceflight (Δ post/preflight), however independent from individual muscle property or condition. By contrast, the “non-exercise” SDM astronaut showed a more severe SOL myofiber CSA decrease, similar to earlier findings reported from short-duration (approximately 7–11 days or more) Space Transportation Shuttle (STS) mission astronauts [48]. Together with the two other LDM astronauts of SARCOLAB [20], the present study outcome provides compelling evidence for the efficacy of inflight exercise protocols for LDM astronauts in spaceflight, although resulting in a variable exercise benefit outcome for the individual mission crewmembers. 

By contrast to the SDM routine, H&E scout histology of post vs. preflight LDM sample cryosections also showed changes in intramuscular connective tissue proportions characterized by expanded parenchymal “gaps” as result of the possible disintegration of intramuscular ECM network molecular components as part of the endomysial sheaths that otherwise provide structural as well as mechanical support for functional myofiber integrity in normal human skeletal muscle. The turnover of ECM molecules, in particular intramuscular connective tissue components, is a hallmark of skeletal muscle remodeling, e.g., in development and adaptation [49], but also in normal aging processes [50] or in fibroproliferative diseases [51]. In the postflight LDM cryosections from the small biopsies containing selected areas of myofibers together with intramuscular endomysial tissue, we found reduced ECM component immunostaining patterns of two collagen types COL4 (-> network-forming fibrils, basal lamina) and COL6 (-> beaded filaments, myofiber integrity), basement membrane-associated proteoglycan Perlecan (-> linking collagen to dystrophin complex, -> myogenesis, -> fibrosis), and one canonical ECM breakdown-associated matrix metalloproteinase type 2, MMP2 (-> collagen degradation) vs. preflight baseline, altogether suggesting profound molecular adaptation/remodeling mechanisms associated to perimysial/endomysial intramuscular connective tissue during LDM spaceflight. In human muscle, signs of peri/endomysial changes were previously found around atrophic myofibers (MMP-2 accumulation) in inflammatory myopathies [52]. By contrast to the LDM muscle changes reported from the present study, similar tissue changes (e.g., enlarged peri/endomysial gaps) were not reported from the atrophied SOL (unchanged endomysium content relative to myofibers) following bed rest disuse (bed rest-only control samples) as analog to spaceflight [53]. Together with our results, these finding suggest unique yet underexplored intramuscular ECM remodeling mechanisms common to human skeletal muscle adaptation to variable magnitudes of mechanical strain (via inflight exercise) in spaceflight and following bed rest immobilization with apparently low or very low mechanical strain intensities of disused muscle in the absence of CM exercise on Earth, requiring further investigation. 

Proteome analysis of the same samples confirmed dysregulation (loss or gain vs. baseline) of many other structural protein expression levels, including ECM, cytoskeletal, or microtubule organization proteins, which were mainly observed after SDM (e.g., fibrillin-1, FBN1), and sparsely detectable at variance (DPYSL3, PLEC and TUBB4B) among the same protein categories after LDM, suggesting a number of remodeling processes still to occur as signature to more complex molecular adaptation processes in skeletal muscle and myofibers in long-duration spaceflight. Other structural proteins associated to muscle contraction in LDM astronauts only showed trends for a change in proteins associated to muscle development, contractile, sarcomeric, and tubule organisation similar to the SOL muscle proteomic responses found in previous, controlled, 55-day long-term bed rest (LTBR) study participants who underwent resistive muscle vibration exercise (RVE) as an efficient countermeasure and an alternative exercise protocol to combat immobilization-induced disuse atrophy [34].

Although invasive in nature, as a matter of fact, small muscle biopsies are well-tolerated voluntarily by otherwise healthy participants in other spaceflight analogues, such as bed rest [54,55,56], in the critical care patient environment [57], in studies on nutrition/metabolism [58,59] or aging/sarcopenia [60]. Besides, more easily available “liquid” biopsies from relevant biofluids, e.g., urine, blood, or even other probably “less-invasive” sampling procedures (saliva, hair bulbs), via small muscle biopsy samples, are still an expedient approach and the only way to study cell and tissue structure and function changes directly in a given muscle of interest particularly in astronauts before and after spaceflight [61] and to obtain large-scale proteome datasets from human tissue and many other organs, e.g., to be investigated under normal or various extreme environmental conditions. 

### Study Limitations

Some study limitations and open questions should be taken into account. For instance, due to the limited number of human subjects, which is an inherent limitation of space-related research, this work is considered an observational study in terms of basic category or fundamental molecular science study. Nevertheless, studies on a unique population of astronauts likely help to elucidate underestimated yet key mechanisms and proof of concept/hypotheses of microgravity-induced molecular adaptation processes reflected, e.g., by canonical and other muscle-specific biological pathways responding at cellular and tissue levels otherwise masked by gravity on Earth. Another study limitation is given by the comparison of datasets available from one (n = 1) SDM astronaut with those available from four (n = 4) LDM astronauts, thus limiting statistical strength between group comparison. However proteome datasets available from an SDM astronaut as a “non-exercise” inflight control subject (w/o inflight CM exercise) are nevertheless more than precious if compared to the same datasets obtained by identical methodological and analytical approaches from several LDM astronauts (performing routine inflight CM exercise) to highlight key changes in multiple muscle parameters (structural vs. omics), to gain deeper insights in human muscle adaptation and plasticity, but also in terms of lessons learned from inflight exercise outcomes in spaceflight in otherwise healthy, well-trained, and mostly fit astronauts following acute vs. chronic microgravity exposition in still exploratory space missions. The multiple signs of intramuscular ECM remodelling found both in cryostat sections (both histology and immunohistology) and in the structural ECM organization protein expression patterns in LDM despite their routine inflight exercise CM (weekly onboard exercise regimen) is a remarkable and new finding that needs to be better understood. Finally, only male participants enrolled for the Muscle Biopsy experiment and thus we were not able to study sex/gender differences between astronauts in spaceflight.

## 4. Materials and Methods

### 4.1. Study Design

This study is based on human skeletal muscle biopsy material (pre- vs. post-flight) collected for the European Space Agency (ESA) solicited spaceflight experiment (ESA op.nom. Muscle Biopsy) with ISS astronauts between the years 2015 to 2019. Here, we report the results from structural and proteome analyses of soleus (SOL) muscle biopsies obtained from four (n = 4) male long duration mission (LDM, 6 months/180 days or more) astronauts of comparable body mass index (BMI, kg x m^−2^) referred to as Astronauts A–D (Table 1). The study also includes SOL muscle biopsy data relevant for one (n = 1) short duration mission (SDM) astronaut (total 11 flight days including 9 days onboard ISS). All study participants signed an Informed Consent Form, Multilateral Human Research Board (HRMRB) document, including Subject Information Handout and Laymen’s Summary, provided and approved by the Institutional Review Board (IRB) of the National Aeronautics & Space Administration (NASA), Japanese Space Agency (JAXA), ESA Medical Board, and by the Charité—Universitätsmedizin Berlin Ethics Board (verdict EA4/057/08), Germany, before study inclusion. The study protocol and all measurements and procedures complied with the Declaration of Helsinki (54th Revision 2008, Korea) on the treatment of human subjects. Participants were allowed to withdraw from the study at any time.

### 4.2. Muscle Biopsy

Four LDM astronauts (A to D) and one SDM astronaut (SDM) gave a small muscle biopsy (approximately 50 to 75 mg) from their left leg deep calf soleus muscle (SOL) during a baseline data collection session (BDC) before launch (preflight BDC-90 ± 30 days), and shortly after landing within 24 h after return (postflight R±0/1) back on Earth. Biopsies were performed under sterile conditions and with local anesthesia (1% Lidocaine^®^). After incision into skin and fascia, an established Rongeur forceps (conchotome) protocol previously reported for bed rest was used [62]. Fresh tissue samples were separated in smaller pieces for histology/immunohistology or protein analyses (proteomics), immediately frozen in liquid nitrogene (fl N_2_) on site, shipped in deep frozen state, and stored at −80 °C in a laboratory freezer until use. All biopsies were harvested under full medical wound care by two well-experienced experts from the science team (J.R., U.L.) at two different locations, either at the European Astronaut Center (EAC) with nearby Human Physiology Laboratory of the German Space Agency (DLR), Cologne, Germany, or at NASA’s Johnson Space Center (JSC), Houston, Texas, depending on astronauts’ travel plans and operational landing conditions.

### 4.3. Inflight Countermeasures (CM)

During their stay on the ISS, one astronaut (SDM) did not perform routine onboard CM physical exercises due to the relatively short mission duration (9 ISS days and 11 flight days in total). LDM astronauts A–D performed up to 2.5 h of daily physical countermeasure (CM) using approved CM exercise devices (running, cycling, weight-lifting) [22] mainly addressing heart and cardiovascular system (T2 Treadmill/Cycle Ergometer with Vibration Isolation and Stabilization, CEVIS) or muscle and bone system (Advanced Resistance Exercise Device, ARED) as prescribed by the Medical Operation (MedOPs) teams of their affiliated Space Agencies [22,63,64]. The total number of onboard exercise ISS mission days for the different CM devices were documented for LDM crew participants to the study (Table 2). 

### 4.4. Histology and Immunohistochemistry

For histology, cryostat sections (LEICA CM-2800, 8 microns [µm] thickness) were stained by conventional hematoxilin-eosin (H&E) to obtain histologic scout overviews from each biopsy sample. For myofiber size and phenotype, an immunofluorescence staining protocol established for human skeletal muscle tissue was used for slow (type 1) and fast type (type 2) myofiber size determination (cross-sectional area, CSA, given in µm^2^) on 10 microns (µm) serially frozen-cut and 4% buffered paraformaldehyde-fixed cryosections (CM1860, LEICA, Wetzlar, Germany). 

For ECM immune analyses, cryosections were fixed (1:1 ethanol/aceton, 10 min, −20 °C), rinsed in buffer, and preincubated with mouse IgG blocking reagent (1:500 M.O.M., Vectorlabs) in buffer. ECM proteins were analyzed by primary anti-collagen type 4 (α-chain rabbit monoclonal. 1:2000, Abcam ab 199720), anti-collagen type 6 (mouse monoclonal, 1:2000, Abcam ab6311), perlecan (rat IgG2a, 1:400, Invitrogen MA-06821), and matrix metalloproteinase 2 (1:100), with secondary antibodies: Alexa 488, 555 and 635-conjugated affinity purified goat anti-mouse/anti rabbit secondary antibodies (Invitrogen Inc., 1:4000, Carlsbad, CA, USA). The Wilcoxon signed rank test was used in all cases with *p* ≤ 0.05 significance level. We used SPSS (v.25, IBM, Armonk, NY, USA). 

To balance inter-subject variability, subject-matched (post- vs. pre-flight) cryosections were always incubated in parallel always at the same time with the same immunohistochemical protocol.

A high-resolution and three-channel confocal laser scan microscope (SP-8, LEICA Microsystems, Germany) was used as previously described [65]. Microscope images were further processed by built-in Leica Application Suite (LAS)-X co-localization/3D image analysis software (release # 3.5.7.23225.3D). GraphPad Prism (v 9.4.0) was used for graphical presentations of statistical data analyzed from at least three (pre vs. postflight) different cryosections from pre vs. postflight astronaut biopsies, one SDM and three LDM astronauts A–C (ANOVA, post hoc analysis, paired Student’s *t*-test/Dunn’s multiple analysis test, at *p* ≤ 0.05 ± standard deviation, S.D.). Unfortunately, inadequate myofiber orientation in one pair of LDM biopsies (astronaut D) did not allow for subject-matched morphometry and histology/immunohistochemistry analyses.

### 4.5. Protein Extraction for Label-Free LC–ESI–MS/MS Analysis

Pre- and post-flight muscle biopsies (SOL aliquots) were suspended in HEN buffer (100 mM HEPES pH 7,4, 1 mM EDTA, 0,1 mM Neocuproina) 0.1% SDS, 1% TRITON X-100, 1% phenylmethanesulfonyl fluoride (PMSF) and sonicated on ice until completely dissolved. Lysates were clarified by centrifugation at 12,000× *g* for 20 min at 4 °C. Protein quantitation with a Pierce bicinchoninic acid (BCA) protein assay (Thermo Fisher Scientific, Rodano, Italy) was then performed. 

### 4.6. Label-Free Liquid Chromatography with Tandem Mass Spectrometry

Protein extracts (200 µg for each sample) were processed following the filter-aided sample preparation (FASP) protocol [66]. Peptide samples were concentrated, and separated on a Dionex UltiMate 3000 HPLC System with an Easy Spray PepMap RSLC C18 column (250 mm, internal diameter of 75 µm) (Thermo Fisher Scientific, Rodano, Italy), and electrosprayed into an Orbitrap Fusion Tribrid (Thermo Fisher Scientific, Rodano, Italy) mass spectrometer, as previously described [67]. Three technical replicates for each sample were acquired. Mass spectra were analyzed using MaxQuant software (Max-Planck-Institute of Biochemistry, Munich, Germany, version 1.6.3.3) [68]. The maximum allowed mass deviation was set to 6 ppm for monoisotopic precursor ions and 0.5 Da for MS/MS peaks. Enzyme specificity was set to trypsin/P, and a maximum of two missed cleavages was allowed. Carbamidomethylation was set as a fixed modification, while N-terminal acetylation and methionine oxidation were set as variable modifications. Spectra were searched by the Andromeda search engine against the Homo sapiens Uniprot UP000005640 sequence database (78.120 proteins, release 7 March 2021). Protein identification required at least one unique or razor peptide per protein group. Quantification in MaxQuant was performed using the built-in extracted ion chromatogram (XIC)-based label-free quantification (LFQ) algorithm using fast LFQ [69]. The required FDR was set to 1% at the peptide, 1% at the protein, and 1% at the site-modification level, and the minimum required peptide length was set to 7 amino acids. Statistical analyses were performed using the Perseus software (Max Planck Institute of Biochemistry, Munich, Germany, version 1.4.0.6). Statistical analyses were performed using the Perseus software (v.1.4.0.6, Max Planck Institute of Biochemistry, Martinsried, Germany) [70]. For each experimental group, proteins identified in at least 80% of samples were considered. For statistical analysis, paired Student’s *t*-test with a *p*-value threshold of 0.05 was applied, and results revealed the variation of protein expression between R+0 vs. PRE SDM and R+0 vs. PRE LDM. False positives were excluded utilizing the Benjamini–Hochberg false discovery rate test. 

### 4.7. Ingenuity Pathway Analysis

Functional and network analyses of statistically significant protein expression changes were performed through Ingenuity Pathway Analysis (IPA) software (Qiagen, Hilden, Germany). In brief, data sets with protein identifiers, statistical test *p*-values, and fold change values calculated from label-free LC-ESI-MS/MS were analyzed by IPA. The “core analysis” function was used for data interpretation through the analysis of biological processes, canonical pathways, diseases, and bio functions enriched with proteins differentially regulated. Then, the “comparison analysis” function was used to visualize and identify significant proteins or regulators across experimental conditions. *p*-values were calculated using a right-tailed Fisher’s exact test. Activation z-score was used to predict the activation/inhibition of a pathway/function/diseases and bio functions [71]. A Fisher’s exact test *p*-value < 0.05 and a z-score ≤ −2 and ≥2, taking into account the directionality of the observed effects, were considered statistically significant. 

## 5. Conclusions

The present study shows evidence for multiple intramuscular ECM adaptation in astronaut skeletal muscle during long-duration mission (LDM) spaceflight and provides a novel space omics database on acute and chronic microgravity-induced molecular changes characterizing individual human muscle responses following short and long-duration mission astronauts of the ISS. Even if intersubject variability of exercise benefit is an apparent hallmark of astronauts’ current onboard CM protocols, the present study gives further evidence to support that physical onboard exercise clearly is an effective countermeasure (CM) currently available for extended human space missions that, after all, is able to at least partially prevent maladaptations focusing on skeletal muscle tissue as the largest organ in the human body and in the deep calf soleus in particular as one of the “executers” for human motion on Earth and in future extraterrestrial expeditions in the next decades, including both robotic and manned spaceflight.

Similar studies in the future are mandatory to investigate adaptation and maladaptation in other functional parts of the human musculoskeletal system (e.g., bone tendons, fascia), for better assessment of the astronaut’s health management and guideline for mission success as recently outlined in more detail [6]. This would allow to at least partly prevent or minimize some of the probably even more demanding challenges, e.g., physiological and medical risks faced by space travelers/astronauts, for overall mission success in future human deep space (Moon, Mars) and planetary explorations [72].

## Figures and Tables

**Figure 1 ijms-24-04095-f001:**
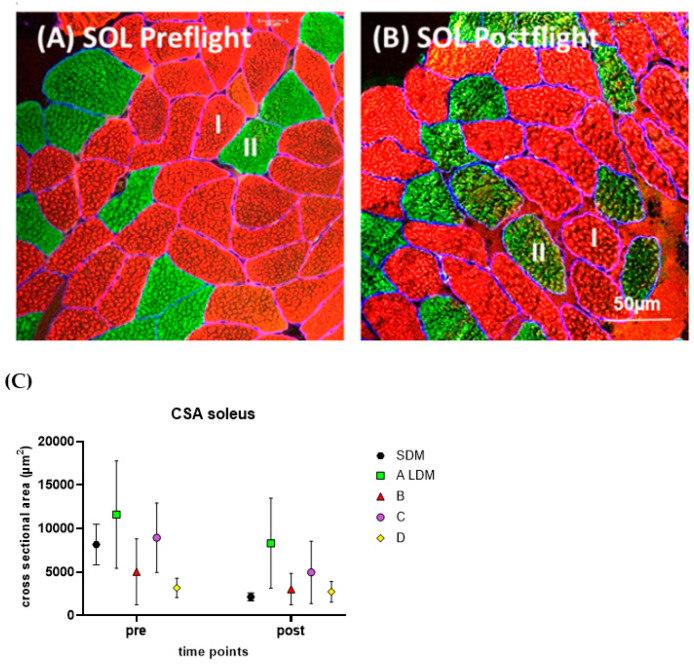
(**A**,**B**) Subject-matched image pair (cross-sectional plane) representative to the SOL from a LDM astronaut (A) immunostained for slow type I (red) and fast-type II (green) myofibers. (**A**) SOL preflight, (**B**) SOL postflight. (**C**) Scatter graph showing mean CSA values of preflight (pre) vs. postflight (post) in SOL myofibers in one short duration astronaut (SDM) vs. four LDM astronauts (A to D). Bar (in (**B**)) = 50 µm.

**Figure 2 ijms-24-04095-f002:**
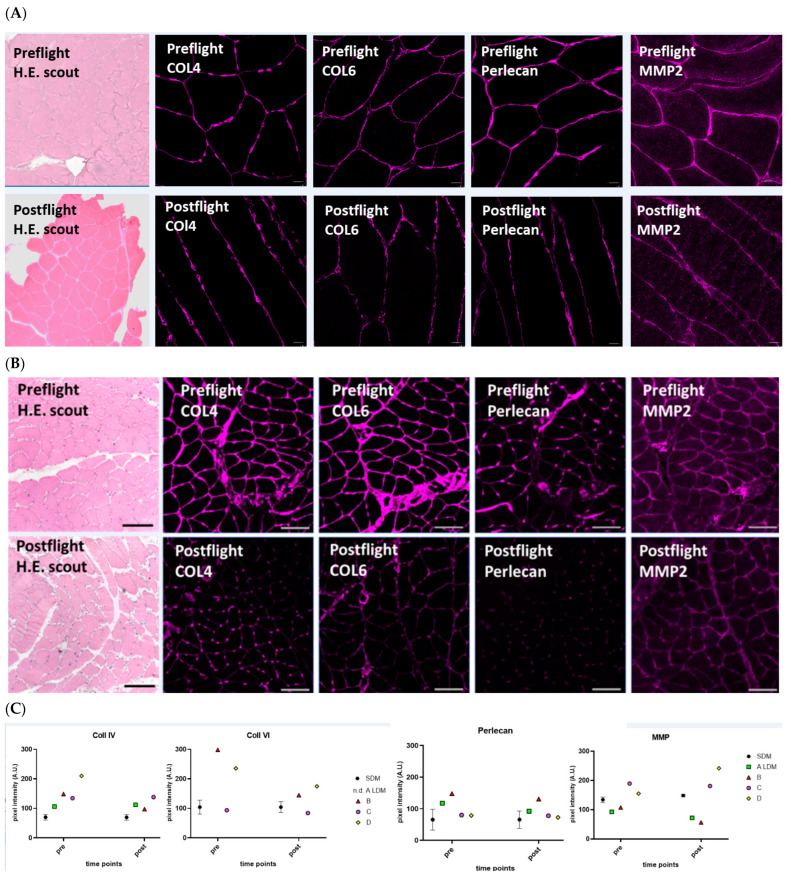
Muscle Histology and Extracellular Matrix Laser Confocal Microscopy Immunosignals analysis SDM and LDM crewmember *SOL* muscle. Subject-matched image (pre/postflight) pairs (cross-sectional plane) representative to the *SOL* from (**A**) short-duration mission crewmember (SDM), (**B**) from LDM crewmember (A) analysed for hematoxylin-eosin (H.E.) scout overview, and immunostained for collagen type 4 (COL4), type 6 (COL6), perlecan, and matrix metalloproteinase-2 (MMP2). Upper panel (preflight), lower panel (postflight). Bars = 50 µm. (**C**) Scatter plots: Each symbol represents ECM values in LDM crewmembers A–D at postflight. Baseline (zero) represents preflight values to highlight signal intensity change (gain or loss) post vs. preflight.

**Figure 3 ijms-24-04095-f003:**
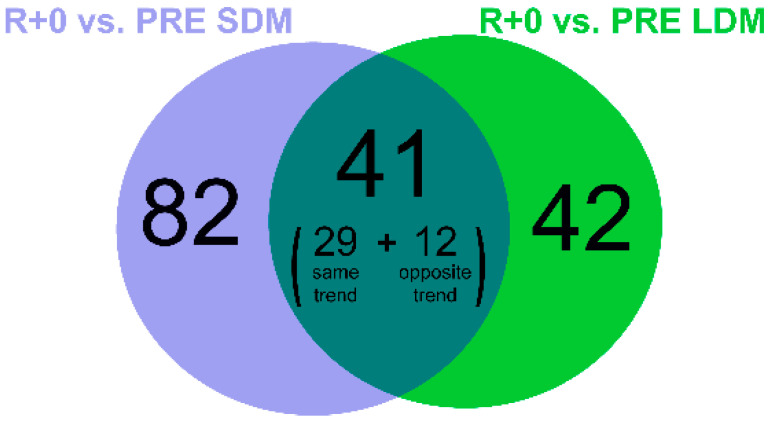
Venn diagram showing the number of identified proteins changed in R+0 (day of return to Earth) vs. PRE (preflight) after 9/11 days (SDM, violet circle) and 6 months (LDM, green circle) in space, as detected with label-free LC-ESI-MS/MS.

**Figure 4 ijms-24-04095-f004:**
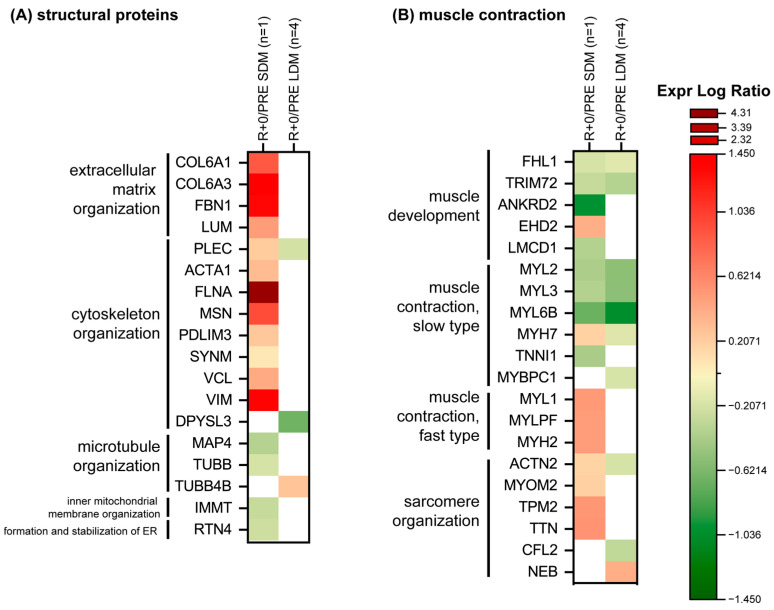
Heatmap of proteins levels divided according to functional categories, (**A**) structural proteins, (**B**) muscle contraction. Green and red colors refer to statistically significant protein decrease or increase (R+0 vs. PRE SDM, paired Student’s *t*-test and FDR, n = 1, *p* < 0.05; R+0 vs. PRE LDM, paired Student’s *t*-test and FDR, n = 4, *p* < 0.05) from the proteomics datasets.

**Figure 5 ijms-24-04095-f005:**
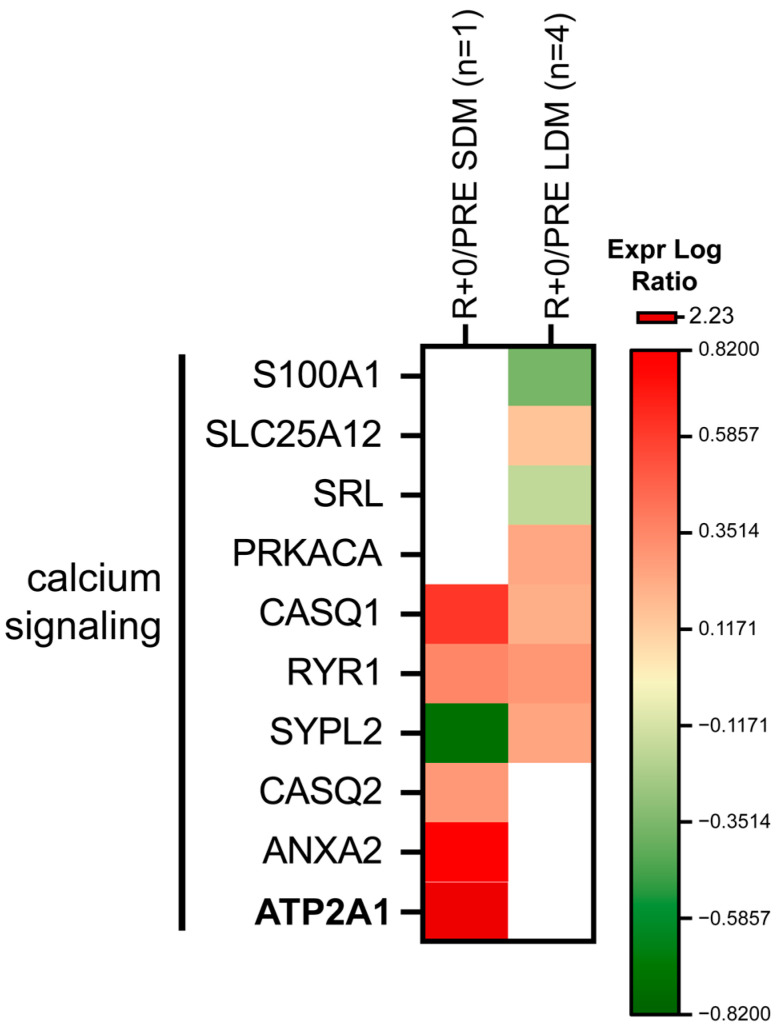
Heatmap of calcium signaling protein levels. Green and red colors refer to statistically significant decrease or increase (R+0 vs. PRE SDM, paired Student’s *t*-test and FDR, n = 1, *p* < 0.05; R+0 vs. PRE LDM, paired Student’s *t*-test and FDR, n = 4, *p* < 0.05) in the proteomics datasets.

**Figure 6 ijms-24-04095-f006:**
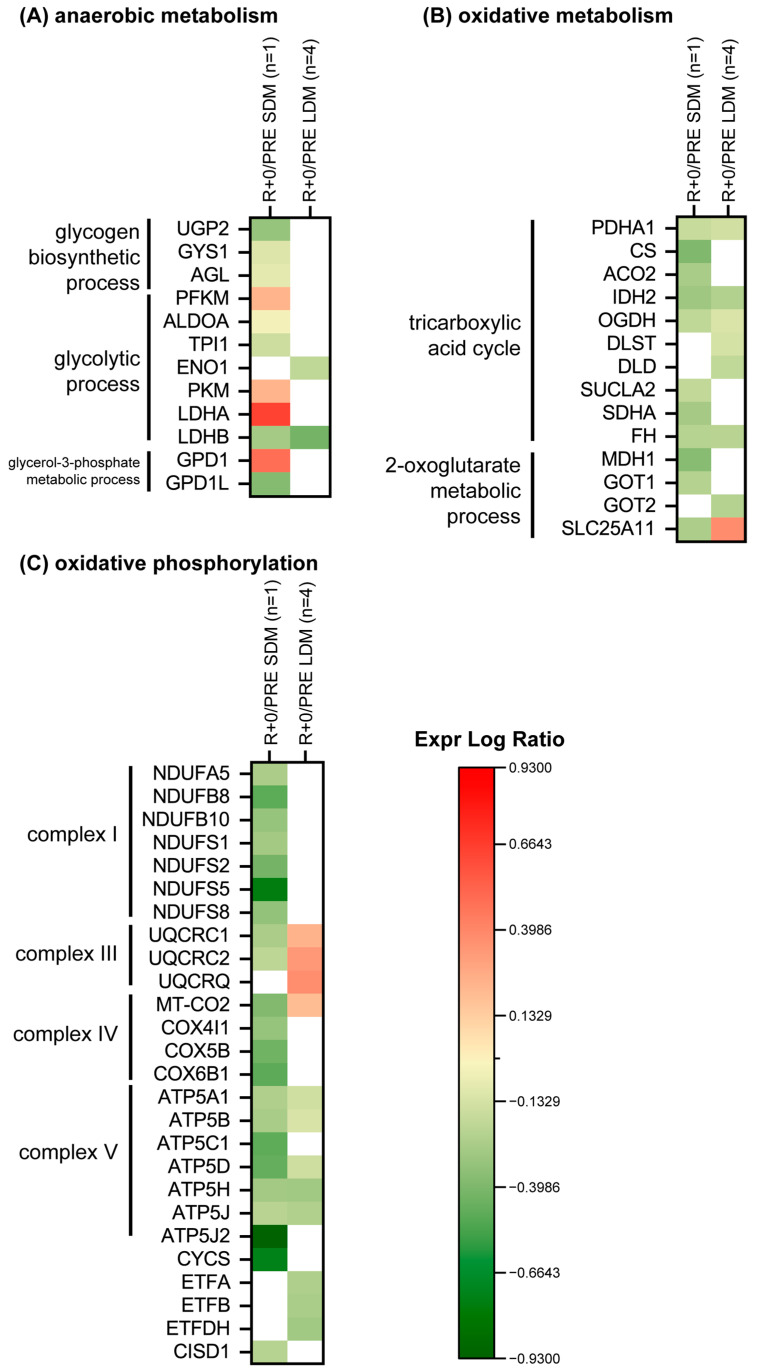
Heatmap of protein levels classified according to a functional category. (**A**) anaerobic metabolism, (**B**) oxidative metabolism, (**C**) oxidative phosphorylation. Green and red colours refer to statistically significant decrease or increase (R+0 vs. PRE SDM, paired Student’s *t*-test and FDR, n = 1, *p* < 0.05; R+0 vs. PRE LDM, paired Student’s *t*-test and FDR, n = 4, *p* < 0.05) in the proteomics datasets.

**Figure 7 ijms-24-04095-f007:**
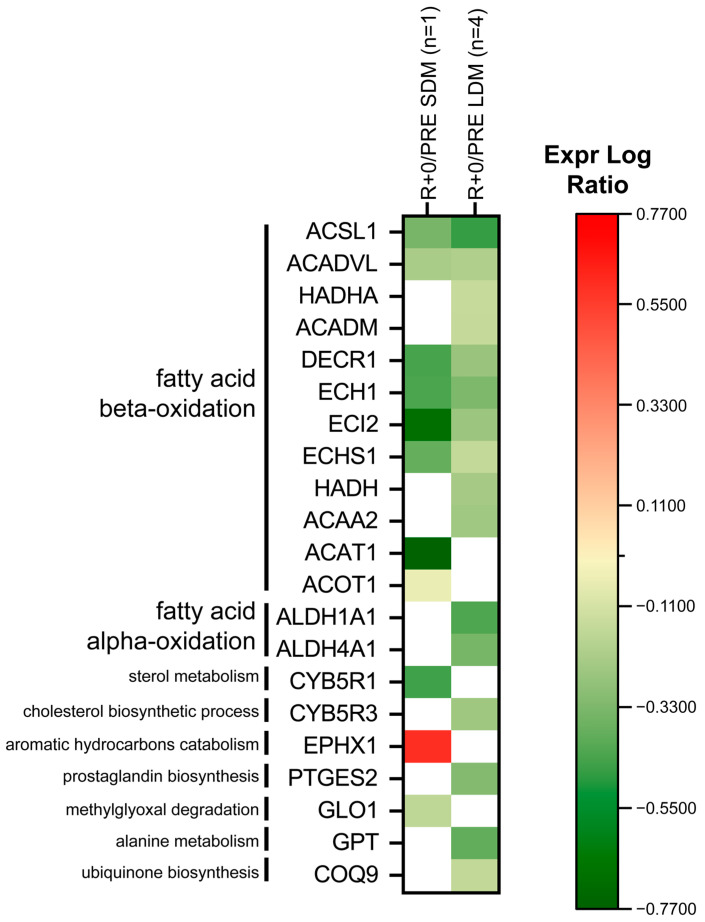
Heatmap of lipid metabolism protein levels. Green and red colors refer to statistically significant decrease or increase for each protein (R+0 vs. PRE SDM, paired Student’s *t*-test and FDR, n = 1, *p* < 0.05; R+0 vs. PRE LDM, paired Student’s *t*-test and FDR, n = 4, *p* < 0.05) in our proteomics datasets.

**Figure 8 ijms-24-04095-f008:**
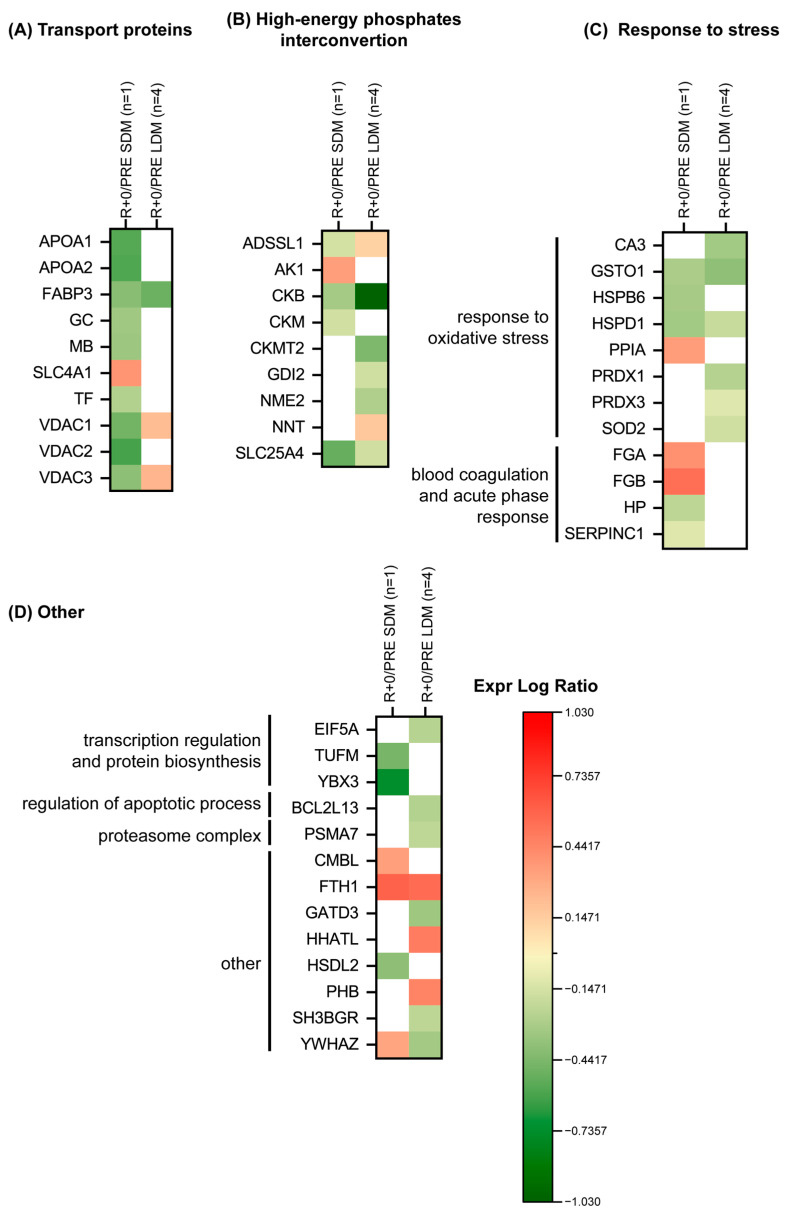
Heatmap of protein levels classified according to a functional category, (**A**) transport proteins, (**B**) high-energy phosphates interconvertion, (**C**) response to stress, (**D**) other Green and red colors refer to statistically significant decrease or increase for each protein (R+0 vs. PRE SDM, paired Student’s *t*-test and FDR, n = 1, *p* < 0.05; R+0 vs. PRE LDM, paired Student’s *t*-test and FDR, n = 4, *p* < 0.05) in our proteomics datasets.

**Table 1 ijms-24-04095-t001:** Body Mass Indices (BMIs) from LDM Astronauts (A to D).

Astronaut (LDM)	Body Weight (kg)	Height (cm)	BMI *
**SDM**	89.4	186.0	25.8
**A**	69.6	172.2	23.5
**B**	87.42	184.0	25.8
**C**	89.37	187.97	25.3
**D**	85.51	183.50	25.3

* age not provided on privacy reasons.

**Table 2 ijms-24-04095-t002:** Number of inflight days (days ISS) for LDM astronauts performing routine CM exercise regimen with CEVIS, T2 and ARED device on ISS (up to 2.5 h/daily) *.

Astronaut (LDM)	CEVIS (Days ISS)	T2 (Days ISS)	ARED (Days ISS)
**SDM**	N/A	N/A	N/A
**A**	45 (FD4 to FD184)	88 (FD10 to FD175)	107 (FD8 to FD181)
**B**	44 (FD6 to FD118	75 (FD8 to FD128)	41 (FD8 to FD175)
**C**	30 (FD7 to Fd128)	64 (FD4 to Fd137)	29 (FD8 to FD175)
**D**	40 (FD6 to FD 153)	68 (FD 5 to FD 133)	39 (FD8 to FD 175)

* CEVIS = Cycle ergometer (heart rate/blood circulation); T2 = Treadmill (endurance, walking/running); ARED = Advanced resistive exercise device (weight lifts); FD = flight days onboard ISS (days ISS); SDM = Short-duration mission; LDM = Long-duration mission astronauts A to D; N/A = not applicable (no inflight exercise).

**Table 3 ijms-24-04095-t003:** Heatmap of canonical pathways displaying the most significant results (ordered by decreasing z-scores) resulting from IPA analysis. Orange and blue indicate predicted pathway activation or predicted inhibition, respectively, via the z-score statistic (significant z-scores ≥ 2, ≤−2).

Canonical Pathways	R+0/PRE SDM	R+0/PRE LDM	Molecules
Oxidative Phosphorylation	**−4.69**	−0.333	ATP5F1A, ATP5F1B, ATP5F1C, ATP5F1D, ATP5MF, ATP5PD, ATP5PF, COX4I1, COX5B, COX6B1, CYCS, MT-CO2, NDUFA5, NDUFB10, NDUFB8, NDUFS1, NDUFS2, NDUFS5, NDUFS8, SDHA, UQCRC1, UQCRC2
TCA Cycle II (Eukaryotic)	**−2.646**	**−2**	ACO2, CS, FH, MDH1, OGDH, SDHA, SUCLA2, DLD, DLST
Necroptosis Signaling Pathway	**−2**	N/A	SLC25A4, VDAC1, VDAC2, VDAC3
GP6 Signaling Pathway	**2**	N/A	COL6A1, COL6A3, FGA, FGB
Fatty Acid β-oxidation I	N/A	**−2.646**	ACAA2, ACADM, ACSL1, ECHS1, ECI2, HADH, HADHA
ILK Signaling	0	**−2.449**	ACTN2, CFL2, MYH7, MYL2, MYL3, MYL6B
RHOA Signaling	0	**−2**	CFL2, MYL2, MYL3, MYL6B
Dilated Cardiomyopathy Signaling Pathway	0	**2.236**	MYH7, MYL2, MYL3, MYL6B, PRKACA

## Data Availability

All human datasets (proteomics) analyzed during the current study are not publicly available due to privacy reasons but are available from the corresponding author upon reasonable request.

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
