# Peer review of "Space Omics and Tissue Response in Astronaut Skeletal Muscle after Short and Long Duration Missions"

_ijms, 2023, doi:10.3390/ijms24044095_

Round 1

Reviewer 1 Report

This manuscript described a space omics and tissue response in astronaut skeletal muscle after short and long duration missions. The data analysis, tables and figures are well presented. Overall, the article is certainly of interest to researchers and deserves publication in IJMS in Special Issue: Novel Molecular Approaches to Skeletal Muscle Disease and Disuse 2.0. But I have a few points which should be addressed before publication.  

1.      In Itroduction section there is a lack of the definition of microgravity.

2.      The Introduction would be beneficial if it contained a justification for choosing the soleus muscle from among the antigravity muscles (quadriceps, and muscles of the back).

3.      Description to Figure 2 should be rearanged because on the body picture there is no panel A,B or C.

4.      There is mistake in lane 167 ‘”Figure 3 the Venn diagram indicates 29 proteins changed in LDM and SDM and 12 proteins with opposite trends”.

5.      Discussion about vitamin and nutrient supplementation during the space flight is needed.

6.      It is worth to discuss the impact of the sex/gender on the metabolic responce during and after space flight.

Author Response

Thank you very much for the valuable suggestions, see below:

1, Please see definition of microgravity in Introduction, line 46 onwards, (µG now also listed in Abbreviations, line 817)

2. Justification for soleus as antigravity muscle: now given in Introduction, line 79, "... one typical postural deep calf leg muscle among other anti-gravity muscles of the human body (e.g. quadriceps, musclers of the back) ..." Please also see Discussion, line 504-513, for further biopsy justification.

3. Fig. 2 re-arrangement with subheadings A) line 148; B) line 150; and C) line 152. See also minor corrections for better reading in Figure 2 legend, see lines 160-161

4. Correction Fig.3. "In Figure 3 the Venn diagram indicates 29 proteins changed in LDM and SDM (same trend) and 12 proteins with opposite trends." 

5. Vitamins and nutrients, see line 432 "... including for example the impact of vitamins and other nutritional interventions related to spaceflight [references]."

6. Sex/Gender discussion, see remarks in Study limitations: lines 538-539

Minor changes from corresponding author:

line 585, "using" removed (double)

Acknowledgements: Martin Gutsmann (Charité Berlin) added (line696)

Author Contributions: Cecilia Gelfi (Milano) added to Conceptualization (line 700)

New reference list included (more citations included for nutrition)

Reviewer 2 Report

REFEREE’S REPORT – COMMENTS TO THE AUTHORS

Blottner et al.:

Paper ID: ijms- 2210046

Paper Title: "Space omics and tissue response in astronaut skeletal muscle after short and long duration missions”

This is a well written manuscript (MS), clearly stating the overall objectives of a well designed and methodologically fully sound study on structural and proteome analyses of deep calf muscle (M. soleus) biopsies obtained from five astronauts pre- and post-flight. Four astronauts were on long duration orbital missions (around 180 days in space), one on a short duration mission (11 days).

From their extensive results, the authors consequently conclude (1) that their study provides a novel space omics database on “…molecular changes [of]…human muscle…” induced by long-term or short-term microgravity and (2) that the study gives further evidence that physical exercise onboard is an effective countermeasure against muscle loss. Their outlook in their chapter on conclusions clearly outlines further directions of research, particularly regarding demanding challenges of future long-term space travel such as, e.g., to Mars.

Certainly, the study has one severe limitation, which is the very limited number of human subjects. The authors are fully aware of this limitation and account for it in a chapter on “limitations” in their discussion. A low number of samples is an inherent limitation of space-related research. Since the study has a great value in fundamental science, this shortcoming has to be regarded as negligible.

The MS can be published as is.

Author Response

We are very thankful for the valuable criticism and commitment for R2 to our manuscript.